# Phyto-Photodynamic Therapy of Prostate Cancer Cells Mediated by Yemenite ‘Etrog’ Leave Extracts

**DOI:** 10.3390/nu16121820

**Published:** 2024-06-10

**Authors:** Efrat Hochma, Paul Ben Ishai, Michael A. Firer, Refael Minnes

**Affiliations:** 1Department of Physics, Ariel University, Ariel 4070000, Israel; efratho@ariel.ac.il (E.H.); paulbi@ariel.ac.il (P.B.I.); 2Department of Chemical Engineering, Ariel University, Ariel 4070000, Israel; 3Adelson School of Medicine, Ariel University, Ariel 4070000, Israel

**Keywords:** phytochemicals, photodynamic therapy, prostate cancer, polyphenolic acids, flavonoids, antioxidants, oxidative stress, reactive oxygen species, leaf extracts, citron, Etrog

## Abstract

Cancer therapy, from malignant tumor inhibition to cellular eradication treatment, remains a challenge, especially regarding reduced side effects and low energy consumption during treatment. Hence, phytochemicals as cytotoxic sensitizers or photosensitizers deserve special attention. The dark and photo-response of Yemenite ‘Etrog’ leaf extracts applied to prostate PC3 cancer cells is reported here. An XTT cell viability assay along with light microscope observations revealed pronounced cytotoxic activity of the extract for long exposure times of 72 h upon concentrations of 175 μg/mL and 87.5 μg/mL, while phototoxic effect was obtained even at low concentration of 10.93 μg/mL and a short introduction period of 1.5 h. For the longest time incubation of 72 h and for the highest extract concentration of 175 μg/mL, relative cell survival decreased by up to 60% (below the IC_50_). In combined phyto-photodynamic therapy, a reduction of 63% compared to unirradiated controls was obtained. The concentration of extract in cells versus the accumulation time was inversely related to fluorescence emission intensity readings. Extracellular ROS production was also shown. Based on an ATR-FTIR analysis of the powdered leaves and their liquid ethanolic extract, biochemical fingerprints of both polar and non-polar phyto-constituents were identified, thereby suggesting their implementation as phyto-medicine and phyto-photomedicine.

## 1. Introduction

Despite impressive progress in molecular cancer biology, in most cases, the challenge remains to develop effective long-term cancer therapies that maintain quality of life [1,2]. One example is malignant prostate cancer (mPrC), one of the leading causes of cancer-related deaths among men over 65 [2,3,4]. Traditionally, chemotherapy, radiotherapy, and surgery for mPrC have been associated with adverse side effects and poor treatment outcomes [2,3,4]. Thoroughly, radical prostatectomy is a common option only for localized prostate cancer. Its efficacy is often predicted by PSA levels (common diagnostic indicator), Gleason score (determines the aggressiveness of the cancer), and the extent of tumor invasion (ranges from localized to metastatic) [5]. External beam radiation therapy (EBRT) and brachy therapy are used to target cancer cells directly. Success predictors include tumor stage, PSA levels, and patient health [6]. Drugs like docetaxel are used for metastatic castration-resistant prostate cancer. Efficacy depends on hormone therapy responses and patient health [7]. Androgen deprivation therapy (ADT) is effective in advanced or recurrent prostate cancer by reducing androgen levels. Predictors of efficacy include initial response to treatment and metastatic status [8,9]. Sipuleucel-T (Provenge) is an option for mPrC. Its success is predicted by disease extent and PSA levels. Emerging therapies also involve checkpoint inhibitors like pembrolizumab [10]. PARP inhibitors are effective only for patients with specific genetic mutations, such as BRCA mutations. Genetic testing is crucial for predicting their efficacy. Studies highlight the use of PSMA-targeted therapies and other novel agents for advanced prostate cancer [11]. The clinical characteristics of affected patients include those who are generally over 50 years old and who experience urinary difficulties, pelvic discomfort, and, in advanced cases, bone pain and weight loss.

Phyto-extracts are an additional and promising source of cytotoxic drugs with low toxicity towards healthy cells, and they therefore deserve special attention as potential anticancer drugs [12,13].

It has long been recognized that green plants contain components that can be used as herbal medicine, natural oxidant scavengers with therapeutic potential, and exhibit antibacterial, anti-fungal, and antiviral properties [12,13,14,15]. Among all fruits and vegetables, citron or ‘Etrog’ waste including its peel, meat, juice, or seeds, is a rich source of bioactive compounds [14,16,17,18,19]. Flavonoids are a class of polyphenolic compounds prevalent in plants, renowned for their diverse health benefits and biological activities [20]. A schematic representation of their chemical structure is illustrated in Figure 1. As depicted, flavonoids form a conjugated system, characterized by alternating double bonds across two fused aromatic rings and a third orthogonal phenolic ring [20]. The fluorophore part is associated with higher electronic energy levels (lower absorption wavelengths) compared to a solely orthogonal benzene ring. Additionally, the structure includes an oxygen that acts as an electron donor, enhancing the compound’s reactivity due to increased charge density. Here, we investigate the Yemenite citrus medica leaves for their dark and visible-absorbed-photosensitive cytotoxic activity.

The ‘Etrog’, an ancient citrus fruit, shares close botanical ties with the pummelo and mandarin. Over time, hybrid citrus fruits were developed in various countries [21,22,23,24,25]. Citrus peels and extracts are highly consumed in Asian countries, with China being the largest producer. The natural anticancer properties of citron led to its interest in in-depth studies as a possible medical application [21,22,23,24,25].

Photodynamic therapy (PDT) can improve and enhance citron phyto-constituents’ anticancer properties. The use of PDT presents a promising improvement in the cancer therapy field, particularly when using lasers as the light source, resulting in enhanced selectivity towards tumor cells without harming healthy surrounding tissues [1,4,13,26,27]. Moreover, whereas tumor cells can bypass drug therapy via genomic instability, becoming resistant to the treatment [1,4,13], tumor cells appear not to develop resistance to PDT [1,4,13,26,28,29].

The basic principle of PDT is the production of reactive oxygen species (ROS) by a visible or near-infrared light-excited photosensitizer (PS), in the presence of molecular oxygen (Figure 1). Exposure of the PS to light will allow either electrons (type I) or energy (type II) to transfer from the excited singlet or triplet PS to a triplet ground state molecular oxygen or biological substrate.

In detail, the type I pathway reaction involves electron transfer from the excited singlet or triplet states of PS to O_2_, leading to the generation of ROS such as: OH*, H_2_O_2_, and O_2_^−^ (Figure 1). The type II pathway reaction occurs via direct energy transfer between the excited triplet state PS (resulting from singlet excited state intersystem crossing), having electrons with parallel spins acting as radicals, and the triplet ground state oxygen molecule, resulting in highly reactive singlet oxygen (^1^O_2_) production (Figure 1) [1,4,13,26,27,28,29,30,31,32,33,34]. The electrons in the first excited singlet state also have numerous pathways to release the absorbed energy back to their ground state, either by emitting light (fluorescence) or by releasing heat (internal conversion).

Most of the chemically synthesized photosensitizers, such as rose Bengal, methylene blue, toluidine blue, squaraine dyes, and others, lack widespread approval for medical use due to safety concerns [26,27,28,29,30,31,32,33,34,35,36]. Therefore, phyto-photosensitizer mediators are becoming a serious alternative.

To the best of our knowledge, Yemenite ‘Etrog’ leaf extracts have not been investigated as cytotoxic sensitizers or photosensitizers, especially for prostate cancer. We investigated the properties of Yemenite ‘Etrog’ leaf extracts and their interaction with PC3 cancer cells. Extract photophysical properties, biochemical profiles, optimal concentration and accumulation time, and cytotoxic performance are presented here. Moreover, the extract’s cyto-phototoxic activity upon laser excitation at 485 nm through combined phyto-PDT is also shown. The 1,3-diphenyl-isobenzofuran (DPBF) indicator was also utilized to investigate the extract’s capability to produce extracellular singlet oxygen under varying illumination periods. The major findings of the present study are as follows: (1) the cytotoxic activity of Yemenite ‘Etrog’ leaf extracts on prostate PC3 cancer cells, with significant cell survival reduction observed even after long exposure times of 72 h; (2) combined phyto-photodynamic therapy resulted in an enhanced reduction in cell survival, suggesting the efficiency of this treatment approach in enhancing the cytotoxic effects of the leaf extracts; and (3) ATR-FTIR analysis identified functional group markers of both polar and non-polar phyto-constituents in the leaf extracts, offering insight into their potential implementation as phyto- and phyto-photomedicines.

Phyto-extracts exhibiting dark and photodynamic activity can be applied as therapeutic and mediation-therapeutic agents in the medical field.

## 2. Materials and Methods

### 2.1. Cancer Cell Line Culture

Human prostate PC3 (ATCC-CRL-1435) cancer cells were cultured at 37 °C in 5% CO_2_ atmosphere in RPMI-1640 (SARTORIUS, Kibbutz Beit Haemek, Israel) medium supplemented with 10% heat inactivated fetal bovine serum (FBS, Biological Industries, Beit Haemek, Israel), 100 U/mL penicillin (SARTORIUS, Kibbutz Beit Haemek, Israel), 100 µg/mL streptomycin (SARTORIUS, Kibbutz Beit Haemek, Israel), and 5% L-glutamine (Sigma, Jerusalem, Israel). Human prostate PC3 cancer cells serve as a valuable tool for investigating potential anti-metastatic therapies, especially bone metastasis, and for evaluating potential therapies targeting metastatic disease. The cells were cultured in T-25 Falcon flasks until 80% confluency was reached. The cells were then de-attached from the culture flasks using 0.05% trypsin 0.02%-EDTA and centrifuged at 500× *g*, for 5 min at 4 °C. Supernatant was replaced with fresh medium, and cells were uniformly dispersed. Cell number and viability were determined with a TC 20^TM^ Automated Cell Counter (BIO-RAD, Tel-Aviv, Israel). For experiments, cells were seeded in 96-well plates at a density of 2 × 10^5^ cells/mL.

### 2.2. Plant Extraction and Stock Preparation

Yemenite citrus medica (‘Etrog’) leaves were collected during September 2023 from around the city of Ariel in central Israel and washed twice with double-distilled water. The leaves were dried at 80 °C for 4 h. Using a coffee bean mill (BOSCH TSM6A011W), the leaves were ground into a fine powder, and incubated at room temperature in absolute solvent ethanol (Bio-Lab, Jerusalem, Israel) at a ratio of 1:10 (gr: mL) overnight while stirring. The extract was filtered twice with a 0.1 µm PVDF membrane filter, and the stock concentration was determined as 3.5 ± 0.35 mg/mL using a rotary evaporator. The stock solution was subjected to a serial dilution with RPMI-1640 culture medium. Working concentrations did not exceed 0.3% *v*/*v* ethanol.

### 2.3. Extract’s Photophysical Absorbance and Emission Spectra

The extract absorbance spectrum was measured at room temperature with a V-730 UV–visible spectrophotometer (Jasco, Hachioji, Japan) in the range of 300 nm to 800 nm, using a 10 mm pathlength quartz cuvette, and a scan rate of 1000 nm/min, where ethanol solvent was taken as the reference. The fluorescence spectra of the extracts were also measured in ethanol with a FP-8500 spectrofluorometer (Jasco, Japan) under the following conditions: excitation wavelength of 485 nm, 500–800 nm measurement range, excitation and emission bandwidths of 10 nm and 5 nm, response time 0.1 s, medium sensitivity, 1 nm data intervals, and 1000 nm/min scan speed.

### 2.4. Attenuated Total Reflection–Fourier Transform Infrared Spectroscopy (ATR-FTIR)

An FT/IR-6800 Spectrometer (Jasco, Japan) was used to measure the mid-IR spectrum absorbed through the extract’s leaf powder and their liquid crude ethanolic extract. The samples were pressed into direct contact with the ATR-prism and the beam was then collected by a TGS detector as it exited the crystal. A 45-degree incident angle was used to collect spectra in the wavenumber range of 4000–400 cm^−1^. The single-reflection diamond ATR device has an effective dimension of 1.8 mm in diameter, and its refractive index is 2.42. In the case of the extract powder, air served as the baseline, and for the liquid crude ethanolic extract, 40% *v*/*v* ethanol in distilled water was examined as a baseline (see also Section 3.5 in Results). The ethanol spectrum as a sample was also observed to differentiate from the liquid crude ethanolic extract spectrum.

### 2.5. Cell Survival and Phyto-Sensitizer Dark Toxicity

PC3 cells in a volume of 100 µL and concentration of 2 × 10^5^ cells/mL were seeded in a 96-well plate (Nunclon, Thermo Fisher Scientific, Waltham, MA, USA) overnight, reaching a confluency of 80%. After serial dilution of the extract with RPMI-1640 culture medium (10% *v*/*v*–0.3% *v*/*v*), 100 µL of each dilution was added to triplicate cell cultures, reaching final concentrations of 175 μg/mL, 87.5 μg/mL, 43.75 μg/mL, 21.87 μg/mL, 10.93 μg/mL, and 5.46 μg/mL (5% *v*/*v*–0.15% *v*/*v*). The cells were exposed to the extract for 2 h, 4 h, 24 h, 48 h, or 72 h. Next, the medium was replaced with 100 µL of fresh medium, followed by the addition of 50 µL of XTT solution, according to the manufacture instructions (BI Biological Industries, Beit Haemek, Israel, Catalog No. 20-300-1000). Control wells of only cells, sensitizer, and ethanol solvent, along with only the culture medium as a blank were also measured. Then, the plate was shaken uniformly for 3 h in the dark. Absorbance of the XTT indicator dye within viable cells was read on a microplate reader at 450 nm (with a 580 nm reference wavelength), and relative cell survival was calculated as follows (ratio of treated cells to the untreated cells):Relative Cell Survival = [(OD_S_^450^ − OD_b_^450^) − (OD_S_^580^ − OD_b_^580^)/(OD_C_^450^ − OD_b_^450^) − (OD_C_^580^ − OD_b_^580^)]
where S, C, and b indicate sample, control, and blank, respectively.

### 2.6. The Minimum Accumulation Time of Extract’s Phyto-Agents within Cells

In a black 96-well plate (Greiner Bio-one, Rehovot, Israel), 100 µL of the cells was seeded at a concentration of 2 × 10^5^ cells/mL overnight (reaching 80% confluency). The cells were exposed to the extract at concentrations where relative dark cell survival was higher than 80% (5.46 μg/mL and 10.93 μg/mL). For various incubation periods of time (0 h, 0.5 h, 1 h, 1.5 h, 2 h, 2.5 h, 3 h, 3.5 h, and 4 h), the fluorescence emission intensity was measured under the following conditions: excitation and emission wavelengths of 485 nm and 675 nm, respectively (extract’s maximum excitation and emission wavelengths), as an indication of extract accumulation.

### 2.7. Laser Toxicity

PC3 cancer cells in a volume of 100 µL and concentration of 2 × 10^5^ cells/mL were seeded in a 96-well plate (Nunclon, Thermo Fisher Scientific) overnight, reaching a confluency of 80%. An argon-based laser with an emission wavelength of 485 nm and an average power density of 100 mW/cm^2^ was used to illuminate the cells. To measure the supplied power, the laser was directed at a 45-degree angle using a dielectric mirror before reaching a beam splitter (10–90%, where 10% was used to measure the power in the 400–700 nm range), as shown in Figure 2. Finally, an additional dielectric mirror was used to vertically focus the laser on the well plate. Then, cells were illuminated for different periods of time to find the longest irradiation time that would not decimate the cells.

### 2.8. Phyto-Photosensitizer as A Mediator for Cancer-PDT

For the PDT treatment, an argon-based laser with a diameter of 1 mm and an emission wavelength and average power density of 485 nm and 100 mW/cm^2^, respectively, was used for the extract excitation. Seeded cells at a 100 µL volume and concentration of 2 × 10^5^ cells/mL in a 96-well plate were cultured with the plant extract phyto-photosensitizer at the suitable concentration and time period found (see phyto-sensitizer dark toxicity and minimum accumulation time sections): 10.93 μg/mL and 1 h exposure time. Then, the cells were subjected to laser treatment for 30 min (according to the laser toxicity findings, see previous section). After irradiation, the plate was incubated for 24 h followed by an XTT assay.

### 2.9. Extract’s Singlet Oxygen Quantum Yield Study

The production of extracellular singlet oxygen by the extract was assessed using the singlet oxygen probe, 1,3-diphenyl-isobenzofuran (DPBF) [37]. DPBF was dissolved in methanol to achieve an optical density (OD) of 0.65 at 410 nm in a final volume of 4 mL, then combined with the extract (100 µL of the extract at a concentration of 3.5 ± 0.35 mg/mL, resulting in an OD of 0.3 at 485 nm). These OD ratios ensure a full range of singlet oxygen trapping by DPBF and prevent aggregation bands. DPBF has a strong absorption band at 410 nm, and the efficiency of singlet oxygen scavenging is indicated by the reduction in this band over irradiation time. The extract was diluted in ethanol (100 µL of the extract at a concentration of 3.5 ± 0.35 mg/mL in a final volume of 4 mL) and its absorption spectra were recorded upon illumination, focusing on the 410 nm and 485 nm wavelengths, serving as a control. Due to the overlap of DPBF and extract absorption at 410 nm, this control was crucial to confirm that the decrease in absorption at 410 nm was solely due to DPBF interacting with the singlet oxygen produced by the extract under laser irradiation at 485 nm. Since DPBF does not absorb at 485 nm, monitoring DPBF alone as an additional control was unnecessary. However, measuring DPBF alone was also conducted to demonstrate no emission signals in the range of 400 nm to 450 nm from the laser excitation source. A relative absorption method was used to calculate the singlet oxygen quantum yield (Φ^1^_O2_) of the extract, with Erythrosin B (food colorant E127) as the reference. According to the literature, the singlet oxygen quantum yield of E127 in aqueous media is Φ^1^_O2_^ref^ = 0.62 [38]. Irradiations were carried out with a laser of emission wavelength 485 nm and an average power density of 100 mW/cm^2^ (see also Section 2.8). A plot of the absorbance of DPBF at 410 nm versus time was generated, and the singlet oxygen production rate was obtained from the fitted slope of the pseudo-first-order reaction:ln(C/C_0_) = −kt(1)
where C represents the DPBF concentration at 410 nm over time, C_0_ is the initial concentration of DPBF, and k is the degradation rate constant of DPBF [s^−1^M^−1^]. The singlet oxygen quantum yield was calculated according to the following equation of the Φ^1^_O2_ relative absorption method:(Φ^1^_O2_ A)/k = (Φ^1^_O2_ ^r^ A^r^)/k^r^(2)
where k is the degradation rate of the singlet-oxygen indicator DPBF obtained from the pseudo-first-order reaction slop, and A represents the compound absorption at the excitation light wavelength.

### 2.10. Statistics

Each measurement was repeated at least three times in well triplicates. The results obtained from at least three independent experiments were analyzed by a *t*-test. The difference between the results was considered significant when the *p*-value was less than 0.05. Quantitative results are presented as the mean ± standard deviation (STDEV).

## 3. Results and Discussion

In this work, we studied and analyzed the properties of Yemenite ‘Etrog’ leaf extracts, and its interaction with prostate PC3 cancer cells. To achieve these goals, the extract’s photophysical properties such as absorbance and emission profiles, functional group analysis of the powdered leaves, and its liquid crude ethanolic extract’s main constituents were examined through infrared transmittance spectroscopy (ATR-FTIR), as well as cell survival/dark cytotoxicity assay and fluorescence emission intensity readings for optimal working concentrations and accumulation times within the cells. Finally, the phototoxicity performance of the extract upon laser excitation at 485 nm was demonstrated through the combined phyto-PDT. Extracellular ROS production was also examined using the DPBF indicator. The overall investigation steps are presented in the following sections.

### 3.1. Extract’s Visible–Near-IR Absorbance and Emission Spectra

Yemenite ‘Etrog’ leaves were extracted in an ethanol absolute solvent as described in Section 2.2 to investigate the influence of its bioactive compounds on prostate PC3 cancer cells in the dark and further upon photodynamic treatment. According to previous studies [12,39], ethanol and methanol extractions may provide the highest content of total polyphenolic, bioflavonoids, and vitamins such as ascorbic acid. As the extract is practically a bioactive array, our goal was to investigate whether the leaf contains natural cytotoxic phyto-agents and/or natural visible light-absorbing-reactive photosensitizers that can be used safely in the medical field. In line with this, the leaf extract was then measured for its photophysical properties. Figure 3a displays three dominant absorbance signals in the visible-light region between 400 nm and 500 nm (at 420 nm, 440 nm, and 485 nm). The broad width of the overall superposition signal indicates that the leaf contains substantial amounts of at least three types of visible light–photosensitizer compound candidates. Visible and a knee near-infrared emission signals, at wavelengths of 675 nm and 725 nm, respectively, can be observed in the fluorescence spectrum of the extract (Figure 3b), with the excitation wavelength set to 485 nm. This observation further supports the assumption that the compounds present in the extract are suitable candidates as photosensitizers.

Citrus fruit yield, including stem, rhizome, seeds, roots, peel, pulp, and juice, are known to contain abundant flavonoids and phenolic bioactive compounds as secondary metabolites which are part of the plant’s defense system [2,12,17,36,39,40]. In addition, flavonoids and polyphenolic compounds are known to absorb light in the UV region and part of the visible-light region [39,40,41]. Poly-alcohols are directly correlated with substantially elevated ROS levels, inducing intrinsic necrosis and/or apoptosis signaling pathways that result in cell inactivation [2]. Moreover, poly-alcohols tend to dissociate under environmental pH changes. Therefore, the cell’s phyto-extract interaction may increase via targeting receptor molecules that stimulate cell death [2]. Secondary metabolites’ anti-inflammatory, anticancer, antibacterial, antiviral, antidiabetic, and antioxidant traits have also been well described in the literature through cell homeostasis interference [2,29,36,39,40,41,42]. Competitive mechanism pathways of promoting versus neutralizing oxidative stress provide insight into the cytotoxic activity-dependent nature of the leaf extract based on concentration and exposure time [2,12,17,36,39,40,41,42], as will be investigated in the next section.

### 3.2. Cytotoxic Activity of Extract against Prostate Cancer Cells

The next step of the study was to find the highest concentration of extract that does not eradicate cells in the dark. In other words, the goal was to find concentrations of work greater than the IC_50_ (50% viable cells), where ideally more than 80% of the cells remain viable. To achieve this goal, cells were exposed to increasing concentrations of the leaf extract (5.46 μg/mL, 10.93 μg/mL, 21.87 μg/mL, 43.75 μg/mL, 87.5 μg/mL, and 175 μg/mL) for 2 h, 4 h, 24 h, 48 h, and 72 h incubation periods. The extract stimulation revealed a time- and concentration-dependent reduction in relative cell survival as compared to controls (Figure 4). Gradient XTT absorbance profile analysis demonstrated that the leaf extract could eradicate the cells in the dark; 72 h incubation with 87.5 μg/mL and 175 μg/mL extract resulted in 0.41 ± 0.06 and 0.42 ± 0.08 (** *p* < 0.05) relative cell survival, respectively, which is below the IC_50_ (Figure 4a). At 21.87 μg/mL and 43.75 μg/mL, the relative cell survival obtained was 0.74 ± 0.04 and 0.63 ± 0.03, respectively, which is above the IC_50_ but below 80% cell viability (*p*-value = 0.25 and 0.11, respectively). For 5.46 μg/mL and 10.93 μg/mL, both values exceeded 80% cell viability at 1.00 ± 0.03 and 0.99 ± 0.03, respectively (*p*-value of 0.25 and 0.13, respectively) (see Figure 4a).

For 24 h and 48 h incubation, cell survival was above the IC_50_ at the highest concentrations of 87.5 μg/mL and 175 μg/mL (** *p* < 0.05), while cell viability was above 80% at the mid concentrations of 21.87 μg/mL and 43.75 μg/mL (Figure 4b,c); 0.72 ± 0.01 and 0.65 ± 0.00, 0.89 ± 0.01 and 0.80 ± 0.01, and 0.65 ± 0.08 and 0.55 ± 0.08, and 0.95 ± 0.01 and 0.83 ± 0.01, respectively (see Table 1). The lowest concentrations of 5.46 μg/mL and 10.93 μg/mL performed the highest cell survival values (Figure 4b,c) of 0.91 ± 0.01 and 0.93 ± 0.01, and 1.00 ± 0.02 and 0.98 ± 0.03, respectively, as expected (see Table 1).

For short exposure times (2 h and 4 h of extract incubation), both the extract and the ethanol solvent exhibited similar behaviors, showing no toxicity towards cells compared to long-term exposures (24 h, 48 h, and 72 h), where extract stimulation became more dominant (Figure 4d,e vs. Figure 4a–c). Cell survival rates received for 2 h and 4 h incubation were as follows: 0.84 ± 0.10 and 0.75 ± 0.06, and 0.76 ± 0.09 and 0.73 ± 0.08 for the highest extract dosages of 87.5 μg/mL and 175 μg/mL (see Table 1); for the mid as well as for the lowest ones (21.87 μg/mL and 43.75 μg/mL, 5.46 μg/mL and 10.93 μg/mL), the values were as follows: 0.93 ± 0.11 and 0.81 ± 0.07, and 0.95 ± 0.11 and 0.89 ± 0.10, and 0.87 ± 0.11 and 0.81 ± 0.13, and 0.92 ± 0.14 and 0.89 ± 0.07, respectively (see Table 1). The reduction in relative cell survival at the 2 h and 4 h time points showed no statistical significance. In all cases, ethanol alone as a control did not affect cell count, emphasizing the extract’s net cytotoxicity.

Light microscope imaging (see Figure A1 and Figure A2 in Appendix A) also supported the proliferation XTT assay results, revealing changes in cell morphology in a time- and dose-dependent manner. At zero time and without phyto-extract treatment, the cells exhibit rod-like structures (Figure A1a in Appendix A), indicating their adherent, proliferative nature. With increasing phyto-sensitizer concentration, there was more shrinkage, and round cells appeared (Figure A1a–f in Appendix A). At the highest concentration of treatment, only deformed body cells were observed (Figure A1g in Appendix A). Gradient solvent control samples demonstrated that the solvent alone had no influence on cell survival. On the other hand, the extract treatment displayed drastic morphological shape changes after 72 h of cell exposure, as can be seen in Figure A2a–f in Appendix A.

In all time periods analyzed, 5.46 μg/mL and 10.93 μg/mL concentration treatments resulted in an above 80% viability and no morphological changes in the cells. Hence, we decided to proceed with these two concentrations to determine the minimum accumulation time of the extract within the cells, and then to apply and evaluate using phyto-PDT.

### 3.3. An Investigation of the Extract’s Minimum Accumulation Time within Prostate Cancer Cells

Further investigations were conducted to determine the time required for the following concentrations to be exposed to the cells: 5.46 μg/mL and 10.93 μg/mL. Seeking suitable working concentrations correlated with optimal accumulation time.

As shown in Figure 5, fluorescence emission intensity readings were taken between 0 and 4 h. During the incubation period of 0 h to 0.5 h, the fluorescence emission intensity signal increased sharply in both cases of 5.46 mg/mL and 10.93 mg/mL (Figure 5a,b). However, it should be noted that the increase in fluorescence difference between 0 h and 0.5 h in the case of 5.46 μg/mL (Figure 5a) is almost 50% lower compared to the case of 10.93 μg/mL (Figure 5b). Accordingly, for the same time interval (0.5 h) and ata higher accumulation of phyto-agent concentration (almost doubled), an elevated increase in fluorescence emission intensitywas observed. This suggests an opposite trend between phyto-sensitizer concentration and accumulation time. For both the lower concentration (Figure 5a) and the higher concentration (Figure 5b), a constant trend in the intensity was observed between 0.5 h and 4 h of incubation, indicating the end of phyto-sensitizer accumulation.

### 3.4. Photodynamic Activity Induced by Extracts against Prostate Cancer Cells

Finally, the PDT treatment was therefore conducted at a concentration that was compatible with a shorter incubation time (10.93 mg/mL). Producing ROS within the malignant cellular environment may increase the probability of apoptosis pathway stimulation through DNA fragmentation, mitochondrial membrane damage, Golgi damage, etc., resulting in desired tumor cell destruction and death [29,30,31,32,33,34]. The findings of maximum working concentration (10.93 mg/mL) and minimum pre-treatment incubation period (1 h) were essential to perform phyto-FDT treatment. Laser irradiation alone (emission wavelength and average power density of 485 nm and 100 mW, respectively) resulted in a relative cell survival reduction of 25% compared to control (Figure 6). The combined phyto-photodynamic treatment mediated by Yemenite ‘Etrog’ leaf extract demonstrated a decrease in relative cell survival of 63% compared to the unirradiated controls (** *p* < 0.05) (see Figure 6). It should be mentioned that saturation in cell survival was obtained upon 30 min of irradiation time. In other words, the interaction in the contact area between the laser beam (1 mm diameter) and cells in the 96-well plate (1.5 cm diameter) reached its maximum. These results highlight the potential of Yemenite ‘Etrog’ leaf extracts’ phytochemicals as an activation photosensitizer for eradicating prostate cancer cells.

As a result of light treatment, flavonoids and polyphenolic compounds absorb between 400 nm and 500 nm, such as the known Rutin phytochemical and its derivatives, contributing essentially to anticancer photodynamic therapy [36]. Flavonoids and polyphenolic compounds might also function as chelators, strengthening the hydrophilic photosensitizers and hydrophobic cell membrane interaction [12,42,43]. Moreover, upon blue and green light illumination, therapeutic bio-compound synthesis could also be enhanced [42,43,44]. Therefore, the probability of an increase in the formation of therapeutic, neo-therapeutic, and phyto-phototherapeutic agents may increase, potentially leading to cell death.

### 3.5. Investigation of Singlet Oxygen Quantum Yield in Yemenite ‘Etrog’ Leaf Extract

The singlet oxygen quantum yield of the Yemenite ‘Etrog’ leaf extract has not been documented in the literature. Therefore, this study investigated the generation of ROS using DPBF as a singlet oxygen trap [37]. We illuminated the extract and DPBF individually in glass vials, as well as a combination of extract and DPBF (refer to Section 2.9). The results, presented in Figure 7, indicate that neither the extract nor DPBF exhibited changes in their absorption spectra under irradiation (Figure 7a,b). However, the absorption spectrum of the extract/DPBF mixture decreased over time (Figure 7c). These findings clearly demonstrate the extract’s ability to generate extracellular ROS upon irradiation. However, the kinetic curves in Figure 7f (based on Figure 7c,e) indicate that singlet oxygen production upon illumination, which can be described by an exponential decay and fitted to a pseudo-first-order reaction suitable to our model (Section 2.9), was observed only for the first 30 s until saturation. Beyond this point, the reaction transitioned to other pathways. Therefore, considering only the time period compatible with our model, the obtained rate constants for both the extract and the reference photosensitizer E127 are 0.44 min^−1^ and 3.57 min^−1^, respectively. Introducing these rate constants into the following equation of the Φ^1^_O2_ relative absorption method
(Φ^1^_O2_ A)/k = (Φ^1^_O2_ ^r^ A^r^)/k^r^
Φ^1^_O2_ × 0.33)/0.44 = (0.62 × 0.02)/3.57
will provide the extract’s singlet oxygen quantum yield Φ^1^_O2_ = 0.005. This finding concurs within the literature, which reports the singlet oxygen quantum yield of anthocyanins to be in the range of 0.02 to 0.04 [45]. However, it seems inappropriate to consider the obtained value as the singlet oxygen quantum yield of the extract due to the relatively high experimental error.

In conclusion, the production of extracellular singlet oxygen upon the illumination of the extract at 485 nm was observed mostly within less than a minute. Therefore, it was justified to investigate the extract’s minimum accumulation time within cells and apply this to the combined phyto-PDT for intracellular ROS production.

### 3.6. Extract’s Biochemical Fingerprint through ATR-FTIR Spectroscopy

Furthermore, the FTIR absorption spectra of the extract’s leaves and their liquid crude ethanolic extract in the range of 4000 to 400 cm^−1^ as shown in Figure 8 could also provide insight into the extract’s dark and phototoxicity performance. Since the extraction solvent affects the phyto-constituents’ chemical structure and their physical–chemical properties, they may interact with one another and with the extraction solvent as well. Therefore, a calibration curve of the extraction solvent (ethanol) at different concentrations (diluted in distilled water) was examined via absorbance spectroscopy (focusing on 210 nm ethanol maximum absorbance) and compared to the liquid crude ethanolic extract optical density. As a result, 40% *v*/*v* ethanol was taken as the baseline in the case of the liquid ethanolic extract (black line, Figure 8). The powder source spectrum (blue line) was measured by taking air as a baseline. The ethanol spectrum as a sample was also observed (red line) to differentiate it from the liquid crude ethanolic extract spectrum. The objective was to determine whether the extraction solvent was effective at extracting the main bioactive compounds from the leaves and to investigate the biochemical profile of the samples.

The liquid crude ethanolic extract (black line) showed a phase shift in its spectra compared to the powder source (blue line, Figure 8). Additionally, in the range of 3700 cm^−1^ to 3000 cm^−1^, the half maxima widths of the ethanolic extract (centered at 3354 cm^−1^) and for the extract’s powder (centered at 3313 cm^−1^) obtained were 223 cm^−1^ and 483 cm^−1^, respectively, compared to 244 cm^−1^ of ethanol alone (centered at 3330 cm^−1^, red line). The position of the centered signals at 3354 cm^−1^ and 3313 cm^−1^ represents stretch vibrations of hydroxyl groups -OH and/or symmetrical stretch vibrations of amide A -NH groups [46,47,48], giving rise to an increased probability of elevated ROS levels [46,47,48,49,50,51,52]. Through polar substances, hydroxyl groups derived from phenol and alcohol compounds may also infer cytotoxic activity performance. Moreover, according to [44,48,49,50,51,52], these mentioned functional groups may indicate the presence of proteins and fatty acids in the liquid ethanolic extract. Amide frequencies can be obtained from the protein’s α-helix structures [44,45,46,47,48,49,50,51]. Continuing to investigate along the spectra, a knee signal at 3071 cm^−1^ and at 3057 cm^−1^ can be observed for the extract powder source (blue line) and for the ethanolic extract (black line). These signals do not appear in the ethanol solvent alone (red line). Meaning, the presence of terpene molecules in the extract through the stretching vibrations of C-H aliphatic groups [44,45,46,47,48,49,50,51,52] is also optional.

The following signals in the liquid crude ethanolic extract (black line) spectra overlap with the extract powder source (blue line): at 2915 cm^−1^, 1657 cm^−1^, and 1753 cm^−1^. These signals can be attributed to stretching vibrations of the C-H bonds in aliphatic (saturated) hydrocarbons, and to >C=O carbonyl group’s stretch vibrations and N-H bending vibrations [44,45,46,47,48,49,50,51,52], respectively, that can also contribute to reactive oxygen species formation and even act as an electron donor, increasing radical formation. The signal at 1413 cm^−1^ can be seen in both the liquid crude ethanolic extract (black line) and the extract powder (blue line), but not in the ethanol sample (red line), indicating successful extraction from the extract powder source at this wavenumber position. At 1320 cm^−1^, a dominant black amplitude (compared to the red line) in overlap with the blue line spectra can be seen, also suggesting an effective extraction.

The three packed signals between 1000 cm^−1^ and 1100 cm^−1^ in the liquid crude ethanolic extract spectra (black line) have a phase shift with the ethanol spectra (red line) at that region but overlap with the powder source (blue line). This also means that there is a probability that the bioactive herb compounds were successfully extracted from the powder source in that region. The small peak at 1150 cm^−1^ that corresponds to the powder source (blue line) can also be seen in the crude ethanolic extract (black line).

The group of signals in the region between 1500 and 500 cm^−1^ are directly associated with polyphenolic compounds as the major constituents of the extract array [50]. The peaks between 1450 and 1650 cm^−1^ are mostly attributed to -C=C- aromatic ring stretching vibrations [44,45,46,47,48,49,50,51,52]. These aromatic compounds are directly correlated to enhance ROS generation [48]. To conclude, the powdered leaves and crude ethanolic extract showed functional group fingerprints of both polar and non-polar phyto-constituents, giving rise to their use as phyto- and phyto-photomedicines.

## 4. Conclusions

The studied extract demonstrated a time- and concentration-dependent cytotoxic effect against prostate PC3 cancer cells, with 72 h as the most pronounced time-dependent loss of cell survival at the concentrations 175 μg/mL and 87.5 μg/mL. Within 24 h and 48 h of exposure, phyto-extract decreased cell survival to similar levels. In short-term exposure times (2 h and 4 h), the phyto-extract showed no dark toxicity towards cells, whereas in long-term exposures (24 h, 48 h, and 72 h), extract cytotoxicity was more pronounced. The study of minimum accumulation time of the extract’s phyto-agents within cells demonstrated an inverse relationship between phyto-sensitizer concentration and accumulation time. The Yemenite ‘Etrog’ leaf extracts also exhibited a photo-response to photodynamic therapy with an emission wavelength and average power density of 485 nm and 100 mW/cm^2^, respectively. A decrease in relative cell survival of 63% compared to unirradiated controls was observed under conditions where the phyto-sensitizer concentration was 10 μg/mL, with a total treatment time of 1.5 h. The study of singlet oxygen quantum yield demonstrated the extract’s ability to generate extracellular singlet oxygen. An ATR-FTIR study demonstrated that ethanol was found to be an effective solvent for the extraction of bioactive herb compounds from the leaves’ powder. Additionally, the ATR-FTIR analysis suggested the presence of polyphenolic acids, proteins, fatty acids, and terpene molecules, which may contribute to enhanced extract/cell interaction and elevated ROS formation. To conclude, Yemenite ‘Etrog’ leaves’ natural phyto-agents and visible light-absorbing-reactive photosensitizers may be applied medically.

## Figures and Tables

**Figure 1 nutrients-16-01820-f001:**
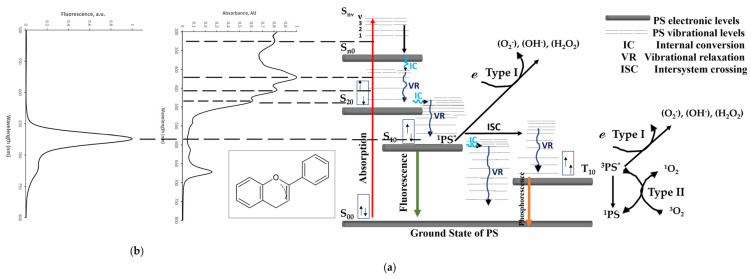
(**a**) Photodynamic therapy mechanism via the Jablonski diagram (inspired by [18]) correlated with (**b**) the extract’s absorption and fluorescence spectra. The subfigure depicts the general chemical structure of Flavonoids.

**Figure 2 nutrients-16-01820-f002:**
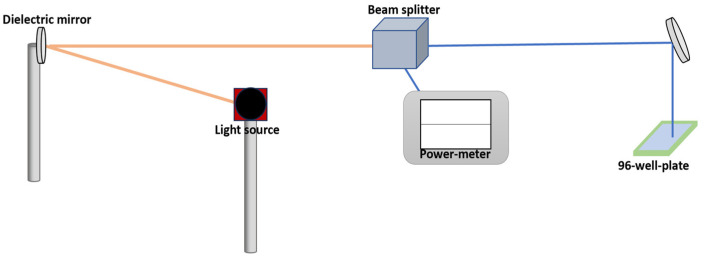
Schematic representation of photodynamic therapy system.

**Figure 3 nutrients-16-01820-f003:**
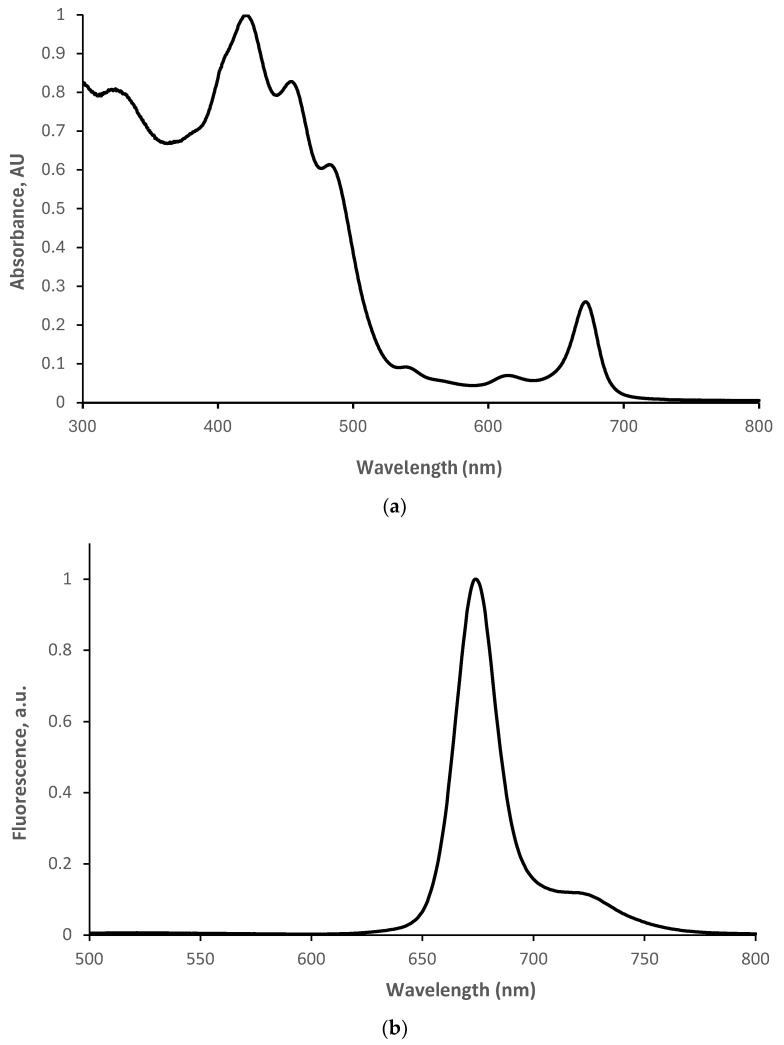
(**a**) Absorbance and (**b**) fluorescence spectra in the range of 300 nm to 800 nm and 500 nm to 800 nm, respectively, of Yemenite citron (‘Etrog’) leaf extracts in ethanol. Slit: 5-5, sensitivity: medium, λ_ex_: 485 nm.

**Figure 4 nutrients-16-01820-f004:**
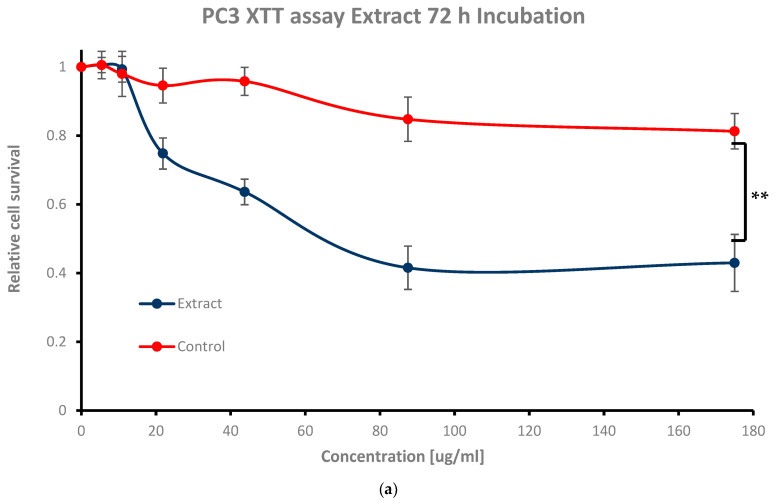
‘Etrog’ leaf extracts’ dark toxicity assay. Cell survival with 5.46 ug/mL, 10.93 ug/mL, 21.87 ug/mL, 43.75 ug/mL, 87.5 ug/mL, and 175 ug/mL of the phyto-extract for (**a**) 72 h, (**b**) 48 h, (**c**) 24 h, (**d**) 4 h, and (**e**) 2 h exposure. Ethanol solvent was taken as a control. Colony survival assay *n* = 3 independent experiments; error bars represent 1 SD; *p*-value < 0.05 (**) was calculated by a *t*-test.

**Figure 5 nutrients-16-01820-f005:**
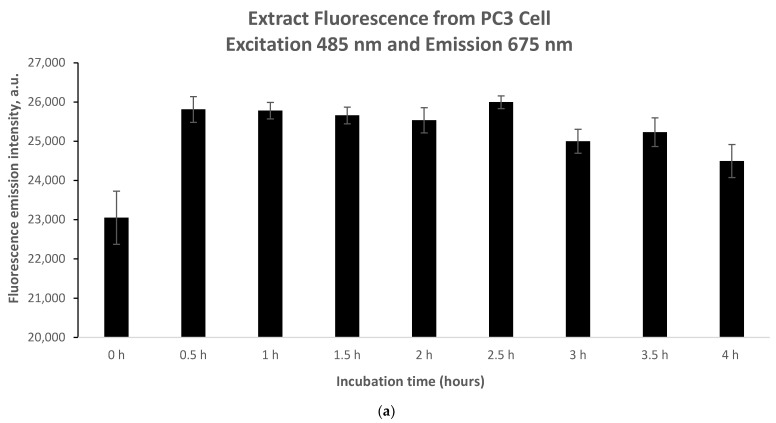
Extract fluorescence emission intensity at 675 nm (upon 485 nm excitation) for (**a**) 5.46 μg/mL and (**b**) 10.93 μg/mL from prostate PC3 cancer cell lines over different incubation times. Data presented as mean and STDEV from three measurements.

**Figure 6 nutrients-16-01820-f006:**
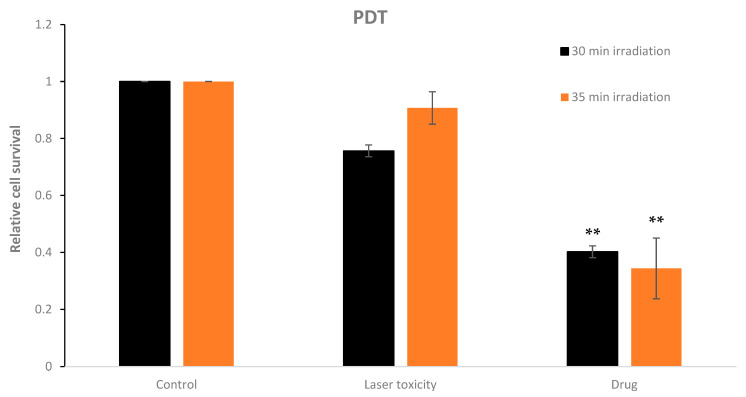
Relative cell survival of prostate PC3 cancer cell line after phyto-photodynamic therapy mediated by Yemenite ‘Etrog’ leaf extract at 10.93 mg/mL concentration and 1 h pre-treatment incubation period. Saturation in cell survival obtained upon 30 min irradiation time. Laser excitation with emission wavelength and average power density of 485 nm and 100 mW/cm^2^, respectively. Data presented as mean and STDEV from 3 measurements. *p*-value < 0.05 (**) was calculated by a *t*-test.

**Figure 7 nutrients-16-01820-f007:**
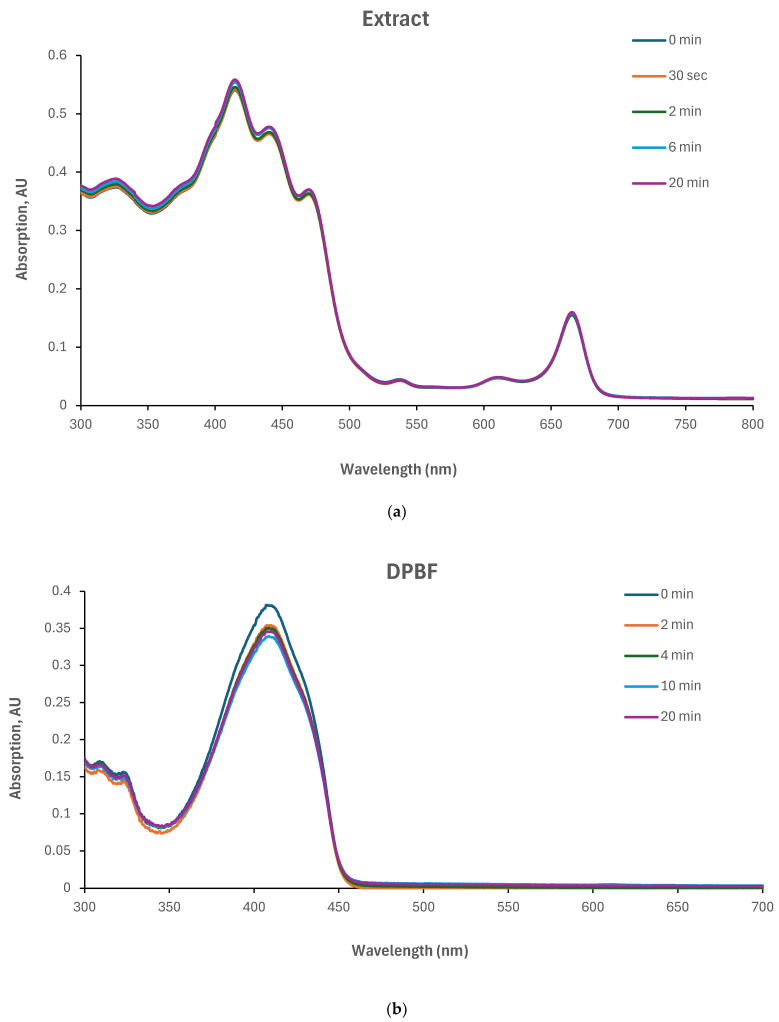
Measuring ROS generation by the extract upon illumination at 485 nm. Absorption spectra of (**a**) extract, (**b**) DPBF, (**c**) DPBF/extract mixture, (**d**) extract and DPBF, (**e**) DPBF/E127 mixture under illumination at different time periods. (**f**) Normalized absorption of DPBF at 410 nm in the presence of the extract and E127 for both short and long exposure times, including their exponential fitting.

**Figure 8 nutrients-16-01820-f008:**
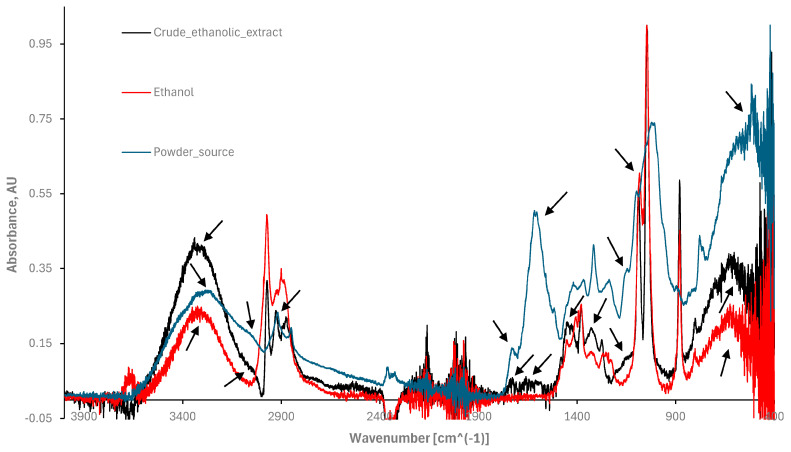
An ATR-FTIR spectrum of the radiation energy absorbed by the leaves’ powder source (blue line, air was taken as the baseline), their liquid crude ethanolic extract (black line correspond to 40% *v*/*v* ethanol as the baseline), and ethanol solvent as a control sample (red line) in the range of 4000 to 400 cm^−1^.

**Table 1 nutrients-16-01820-t001:** Cell survival assay; concentrations and times of exposure to extract.

Exposure Time (hrs.)	Concentration (ug/mL)	Relative Cell Survival_avg	STDEV_avg
0 h	0.00	1.00	0.00
72 h	0.00 5.46 10.93 21.87 43.75 87.5 175	1.00 1.00 0.99 0.74 0.63 0.41 0.42	0.00 0.03 0.03 0.04 0.03 0.06 0.08
48 h	0.00 5.46 10.93 21.87 43.75 87.5 175	1.00 1.00 0.98 0.95 0.83 0.65 0.55	0.00 0.02 0.03 0.01 0.10 0.08 0.08
24 h	0.00 5.46 10.93 21.87 43.75 87.5 175	1.00 0.91 0.93 0.89 0.80 0.72 0.65	0.00 0.01 0.01 0.01 0.01 0.01 0.00
4 h	0.00 5.46 10.93 21.87 43.75 87.5 175	1.00 0.92 0.89 0.87 0.81 0.76 0.73	0.00 0.14 0.07 0.11 0.13 0.09 0.08
2 h	0.00 5.46 10.93 21.87 43.75 87.5 175	1.00 0.95 0.89 0.93 0.81 0.84 0.75	0.00 0.11 0.10 0.11 0.07 0.10 0.06

## Data Availability

The original contributions presented in the study are included in the article, further inquiries can be directed to the corresponding authors.

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
