# Peer review of "Phyto-Photodynamic Therapy of Prostate Cancer Cells Mediated by Yemenite ‘Etrog’ Leave Extracts"

_nutrients, 2024, doi:10.3390/nu16121820_

Round 1

Reviewer 1 Report

Comments and Suggestions for Authors

This is really nice work in the field of Phyto-PDT and may open up new avenues in near future. There are comments below and please try to address them.

1. What would be the chemical structure of Yemenite Etrog leaf extract? How did you characterize the structure other than ATR FT-IR?

2. What about the oxygen concentrations levels of the experiment? 

3. Have you tried the experiments in Hypoxia medium?

4. Have you ever tried to find the generation of singlet oxygen? How about the singlet oxygen quantum yield?

5. Did you any filter to observe the fluorescence experiment? Have you tried different excitation wavelengths?

6. How about the lifetime of this extract in ethanol medium?

7.  Thera are some typos in the manuscripts.

(i) Check page 2, line 49, FDT should be PDT.

(ii) Check page 3, line 117, centrifuged at 500 g. is it 500 rpm?

Author Response

Dear Reviewer,

Thank you for your positive feedback and valuable comments on our work. We have carefully considered each of your suggestions and made the necessary revisions to our manuscript. Below are our detailed responses to your comments:

  1. Chemical Structure Characterization:

The chemical structure of the Yemenite Etrog leaf extract has been further characterized and discussed. In addition to ATR FT-IR, the extract's absorbance and emission spectra were analyzed. Please refer to the updated section 3.1 and the graphical abstract supplied. Additionally, we have included a new paragraph in the introduction (lines 56-64) and updated Figure 1 to provide a clearer depiction of the extract's characteristics.

  1. Oxygen Concentration Levels:

We have added detailed information regarding the oxygen concentration levels used in the experiments. Please see the newly added sections 2.9 and 3.5 for a comprehensive discussion.

  1. Experiments in Hypoxia Medium:

We have not conducted experiments in a hypoxia medium. This is an area for potential future research.

  1. Singlet Oxygen Generation and Quantum Yield:

The generation of singlet oxygen and its quantum yield have been addressed in the newly added sections 2.9 and 3.5. These sections provide detailed methodology and results pertaining to the generation and measurement of singlet oxygen.

  1. Fluorescence Experiment and Excitation Wavelengths:

For the fluorescence experiments, the laser source used emits at 485 nm. Using the DPBF indicator, no other emissions were observed, particularly not in the 400-450 nm range. We chose the least energetic photon in the visible spectrum for excitation based on the extract's absorption spectra. If the lowest energetic photon can induce PDT, higher energy photons will also be effective. Please refer to section 2.9 for further details.

  1. Lifetime of the Extract in Ethanol Medium:

We conducted a photo-degradation experiment to ensure that there was no photo-bleaching of the extract during treatment. Figure 9a illustrates this. Additionally, since the leaves were collected and extracted (section 2.2), their spectral characteristics remained unchanged, indicating the stability of the extract in ethanol medium.

  1. Typographical Errors:

(a) The typo on page 2, line 49 has been corrected from "FDT" to "PDT".

(b) On page 3, line 117, we clarified that the centrifugation was performed at 500 g, not 500 rpm. RCF (relative centrifugal force) is often expressed in units of gravity (g).

We believe these revisions enhance the clarity and comprehensiveness of our manuscript. We appreciate your thorough review and hope that our responses and the revised manuscript meet your approval.

Thank you once again for your constructive feedback.

Sincerely,

Refael Minnes

Reviewer 2 Report

Comments and Suggestions for Authors

The authors should be congratulated for the interesting topic discussed. 

The study has sufficient merit, although major revisions are required. 

1.    Methods and methodology are robust.

2.    Results, conclusions, and limitations are well presented.

3.    These recently published and up-to-date works could enhance the paper's scientific impact: https://doi.org/10.1016/j.critrevonc.2020.102992. The authors should better discuss other therapeutic strategies and their predictor of efficacy and clinical characteristics and presentations of patients affected by PCa

4.    Check the fluency and clarity in some parts of the text.

Comments on the Quality of English Language

Minor editing

Author Response

Dear Reviewer,

Thank you for your encouraging feedback and for highlighting areas where our manuscript could be improved. We have carefully considered your suggestions and made significant revisions to enhance the clarity and scientific impact of our study. Below are our detailed responses to your comments:

  1. Methods and Methodology:

We are grateful for your positive feedback regarding the robustness of our methods and methodology. Your recognition of our approach is greatly appreciated.

  1. Results, Conclusions, and Limitations:

Thank you for acknowledging the presentation of our results, conclusions, and limitations. We have strived to ensure they are clear and well-structured.

  1. Discussion of Other Therapeutic Strategies:

We appreciate your suggestion to include a broader discussion of therapeutic strategies for prostate cancer. In response, we have expanded our discussion to cover various therapeutic approaches, their predictors of efficacy, and the clinical characteristics of patients affected by prostate cancer. These additions are intended to provide a more comprehensive context for our study and enhance its relevance. Please see the newly added lines 35-49 for this expanded discussion. Additionally, the suggested reference (https://doi.org/10.1016/j.critrevonc.2020.102992) has been added to our reference list to further support this discussion.

  1. Fluency and Clarity:

We have reviewed the manuscript thoroughly to improve the fluency and clarity of the text. Specific sections have been revised to ensure a smoother flow and better readability.

We believe these revisions significantly enhance the manuscript, and we are grateful for your constructive feedback. We hope that the changes meet your expectations and improve the overall quality of our work.

Thank you once again for your detailed review and valuable suggestions.

Sincerely,

Refael Minnes

Round 2

Reviewer 2 Report

Comments and Suggestions for Authors

Authors answered all comments and suggestions.

Comments on the Quality of English Language

Minor editing.